# Do Cockatiels Choose Their Favourite Tunes? Use of Touchscreen for Animal Welfare Enhancement and Insights into Musical Preferences

**DOI:** 10.3390/ani14243609

**Published:** 2024-12-14

**Authors:** Mathilde Le Covec, Romain Di Stasi, Carla Aimé, Léa Bouet, Shigeru Watanabe, Dalila Bovet

**Affiliations:** 1Laboratoire Ethologie Cognition Développement, Paris Nanterre University, 92000 Nanterre, France; romain.di-stasi@outlook.com (R.D.S.); mme.carla.aime@gmail.com (C.A.);; 2Centre for Advanced Research on Logic and Sensibility, Keio University, Tokyo 108-0073, Japan; swattaub@gmail.com

**Keywords:** music, welfare, birds, Psittaciformes, Cacatuidae, parrots

## Abstract

In this study, we explored whether cockatiels could learn to use a touchscreen to choose between different pieces of music. Some birds showed individual preferences for either rock and roll or calm music, but no preferences were found for consonance or dissonance. Our findings offer new prospects for the study of musicality in non-humans and its potential applications for promoting the welfare of captive animals.

## 1. Introduction

### 1.1. Musicality in Non-Humans

Music is often considered a human specificity (see, for example [1,2,3] for a detailed discussion). Yet, Darwin, in *The Descent of Man* [4], suggested that a ‘sense of beauty’ (p. 63) might be shared across species, offering an explanation for why we find bird songs appealing. This idea was later supported by studies such as those by Earp and Maney [5], which showed that, in white-throated sparrows (*Zonotrichia albicollis*), conspecific songs activate the brain’s reward pathways in a manner similar to the effect of music on humans.

Musicality (described by Hoeschele and al. [6], as “the capacity that makes it possible for us to perceive, appreciate, and produce music”, p. 1) can be studied in non-human animals at different levels, described by several authors (e.g. [7,8]): (1) the cognitive or discriminative level, i.e., the ability to distinguish between various types of music (such as classical, jazz, and rock and roll) or different composers (such as Bach, Mozart, and Beethoven), and (2) the reinforcing level, i.e., experiencing pleasure when listening to musical stimuli and showing musical preferences. Some studies have already evidenced abilities to discriminate among different rhythms and/or melodies in numerous species, including mammals like elephants *Elephas maximus* [9], rats *Rattus norvegicus* [8,10] and Japanese macaques *Macaca fuscata* [11], birds like jackdaws *Corvus monedula* [12], pigeons *Columba livia* [13], common starlings *Sturnus vulgaris* [14], Java sparrows *Lonchura oryzivora* [15], zebra finches *Taeniopygia guttata* and budgerigars *Melopsittacus undulatus* [16] and fishes like goldfishes *Carassius auratus* [17] and common carps *Cyprinus carpio* [18]. However, a reinforcing effect of music (i.e., perception of music as a pleasant stimulus, shown by a preference for music over silence and/or for one type of music over another) has been found in only a few of these species. In particular, non or limited vocal learners like hens *Gallus gallus domesticus* [19], pigeons [20], or rats [21] do not show musical preferences. Given that they are phylogenetically very close to us, several studies also explored musicality and, more specifically, musical preferences in apes and monkeys, which are limited vocal learners [22]. Those studies show contrasted results: although chimpanzees (*Pan troglodytes*) prefer African and Indian music over silence [23] and one baby female chimpanzee preferred consonance over dissonance [24], marmosets (*Saguinus oedipus*), tamarins (*Callithrix jacchus*) [25], and orangutans (*Pongo abelii*) [26] prefer silence over music. Conversely, in vocal learners, two out of four Java sparrows preferred music over silence and spent more time on perches triggering classical music such as Bach and Vivaldi rather than modern music like Schönberg or Carter [15]; a hyacinth macaw (*Anodorhynchus hyacinthinus*) chose to trigger music by using a joystick or bobbing its head [27], and a study with two grey parrots (*Psittacus erithacus*) showed that, when provided with a touchscreen broadcasting either rhythmic or calm music, the birds displayed individual preferences [28]. However, to date, musical preferences have only been studied in a very limited number of vocal learner species (and with a very small sample size). It would be thus interesting to explore this question further with other species and more data.

### 1.2. The Vocal Learning Hypothesis

For Schusterman [29], vocal learning is an ability consisting of modifying vocal production regarding external auditory information. According to Watanabe et al. [20], musical stimuli would have reinforcing effects mainly in species able to learn complex auditory signals. Thereby, only species with strong vocal learning abilities, like songbirds and Psittaciformes, would show musical preferences. Non or limited vocal learners, on the contrary, would only be able to discriminate musical stimuli.

In particular, Psittaciformes seem to be quite good candidates for the study of musicality in non-humans. Indeed, in previous studies, a grey parrot was able to complex musical production [30], budgerigars synchronised themselves with a metronome [31], and at least six Psittaciformes species spontaneously entrained to a rhythm [32] over a large range of tempi [33]. One of them, a sulphur-crested cockatoo (*Cacatua galerita*), even displayed 14 different movements for that purpose [34]. Additionally, palm cockatoos (*Probosciger aterrimus*) create tools with which they drum on hollow trunks with individual styles [35], and their drumming shares key components with human music [36].

### 1.3. The Current Study

This study is inspired by previous research conducted at our laboratory, exploring musical preferences in two African grey parrots, *Psittacus erithacus* [28]. For this research, a touchscreen connected to a computer was provided to the birds in the aviary. A red square associated with rhythmic music and a dark blue circle associated with calm music were displayed on the screen, allowing the birds to trigger the music by touching one of the shapes. The parrots used the screen even without any food reward and exhibited stable individual preferences. Such a device offers valuable potential for improving animal welfare. Indeed, musical influences on welfare are likely to depend on individual music preferences and individual preferences can be explored thanks to choice tests [37]. Moreover, showing that birds are able to use a touchscreen to choose their preferred sound enrichment is interesting as it shows that touchscreens could be used to increase welfare in captive animals.

Here, we aimed to use a similar device with cockatiels (*Nymphicus hollandicus*) to explore the birds’ preferences in terms of rhythmicity and consonance/dissonance. Cockatiels are *Cacatuidae*, like palm cockatoos. They also belong to the Psittaciforme order, in which birds are capable of vocal learning and known for their complex cognitive skills. Furthermore, their auditory sensitivity runs from 2000 to 5000 Hz [38], which broadly overlaps those of humans (between 20 and 20,000 Hz with increased sensibility from 2000 to 4000 Hz [39]).

We made the following hypotheses:As cockatiels are vocal learners, at least some individuals would have individual music preferences, like in Java sparrows [15] and grey parrots [28].Cockatiels would be able to learn how to use a touchscreen to display their preferred music.Regarding consonance and dissonance, birds would prefer consonance. Indeed, Bowling and Purves [40] suggest that a preference for consonance over dissonance would have evolved as the harmonic series characterising the vocalisations of conspecifics are consonant. Consonant intervals are usually described as smooth or pleasant, whereas dissonant intervals sound rough or unpleasant [41]. Preference for consonance appears very early in humans [41], and even limited vocal learners like chicks [42] and an infant chimpanzee [24] have displayed preferences for consonance over dissonance in previous studies. Additionally, in a previous study (not published yet), we observed that cockatiels behaved differently (i.e., were less aggressive toward conspecifics) when listening to consonant music compared to dissonant music, suggesting that they can discriminate both.

## 2. Materials and Methods

### 2.1. Subjects and Housing Conditions

This experiment was held with twelve cockatiels housed in our laboratory for scientific purposes (see Table 1 for details about the birds). Except Isis, Éole, Morgane, and Gaïa, who were born in the laboratory, all the birds originated from a breeder.

The birds were housed in an indoor aviary with no window and kept under similar housing conditions prior to and during the experiments. Temperature and ventilation were controlled in order to ensure optimal microclimate conditions for the birds (temperature was kept around 23 °C at all times). The light was on from 9 a.m. to 11 p.m., and minimal lighting was used during the night to avoid any night frights. Three stainless steel workbenches (155 × 55 × 84 cm) were located on the right wall, and there were 2 big perches in the middle of the room. Toys, cardboard, and two triangle-suspended perches were also available. Birds were fed ad libitum with, regarding the period, either seed or pellets available on the market for large parakeets, various fruits and vegetables, aniseed sand, and water.

#### 2.1.1. Study 1

A fake touchscreen (25.5 × 18.5 cm), made with a sheet of paper protected by a plastic screen, was used for this study because of technical problems with the touchscreen that was initially supposed to be used. In front of the fake screen, a small platform was set, on which the birds could perch. We drew on this paper screen a red square (10.5 cm × 10.5 cm) and a dark blue circle (radius circle 5.6 cm), both having the same area (110 cm^2^) (see Figure 1). Two music plays were manually triggered by an experimenter when the birds pecked a shape, both broadcasted by a Bluetooth speaker hidden behind the paper screen. The paper screen was visible to the birds only when the experimenter was in the aviary; otherwise, we put cardboard in front of it.

First of all, the birds were familiarised with the screen for two weeks. Millet was permanently placed on the platform and on the screen to encourage them to interact with the device.

Then we conducted 95 one-hour sessions (2 to 4 sessions per day within six weeks, except during the weekends). The birds were free to perch on the platform and participate or not. In order to increase the motivation for the alimentary rewards offered during the sessions’ periods, the food was removed thirty minutes before such trials. Each time a bird touched a shape on the paper screen, the experimenter played the corresponding music and then rewarded the subject with a seed (either sunflower or millet, depending on what they were most motivated to). When there was no interaction with the screen for the first 15 min, the experimenter left the aviary and came back half an hour later to try again. The two music plays broadcasted were 15 s long excerpts of piano versions of *Rock Around the Clock* by Bill Haley (rock and roll music) and *Le roi et l’oiseau* by Wojciech Kilar (calm music). We chose specifically these two music plays because we wanted 2 highly contrasting pieces: Bill Haley is very rhythmic, and the other is calmer. Additionally, both were easy to find performed on the piano. Indeed, we wanted to avoid bias related to instrument preferences by choosing two pieces performed by the same instrument. The training lasted from session 1 to session 28. From session 1 to session 19, the birds were rewarded as soon as they pecked the shape. From session 20 to session 24, we waited for 5 s from the moment the shape was pecked before rewarding the cockatiels to be sure that they listened to the music. From session 25 to session 28, this delay was increased to 10 s. From session 29 to session 45, the birds were taught to wait for the end of the music (i.e., 15 s) before pecking again and were only rewarded by then (condition 1). If one bird wanted to peck while another bird had already started the music, he/she had to wait until the music stopped before pecking; otherwise, he/she was not rewarded. The pass criterion for each bird was a minimum of 300 pecks on the screen and at least 50 pecks on each shape during the training and condition 1. Once these criteria were met, we assumed the birds had understood the setup and moved them to condition 2. Between sessions 46 and 70 (condition 2), the shapes’ positions were switched to check for a position bias. From session 71 to the end (condition 3), the music tracks linked to each shape were interchanged to check whether the birds truly had music preferences or simply favoured a particular shape (see Figure 2 and Table 2 for details).

#### 2.1.2. Study 2

The second experiment was conducted 11 months after the first one, as we tried in the meantime to find a real touchscreen that could be used by cockatiels. Indeed, showing that cockatiels are able to choose and trigger their preferred music by themselves using such a device, without any human intervention, could be more interesting from several considerations: eliminating biases, further use of such devices for EE purposes, etc. Finally, the birds were trained with an Elo 1590L 15-inch LCD screen from Elo Touch Solutions, Knoxville, United States. Two shapes were displayed on the screen. Since the music plays were different from the previous ones, we also changed the shapes because we thought it would be more consistent for the birds and eliminate the risk of habituation: one orange heart and one green cross, having the same area (110 cm^2^) and broadcasting either consonant or dissonant music for 15 s each; these musical stimuli were two different versions of the same Renaissance play, *Entrée courante* (anonymous). The consonant stimulus was the original play, and to create the dissonant stimulus, this original tune was electronically manipulated (using CoolEdit Pro software, by Koelsch et al. [43]), i.e., recorded simultaneously with two pitch-shifted versions, which were one tone above and a tritone below the original pitch. We changed the stimuli for the second experiment to avoid any potential preferences the birds might have had for the previous tracks (study 1) influencing their choice between consonance and dissonance. The birds were supposed to trigger the music by pecking the shapes with their beak but they were still actually not able to activate the music by themselves (probably because the surface of the beak in contact with the screen was too small to be detected by the touchscreen). Therefore, the experimenter systematically touched the shape after the birds to trigger the music.

We conducted 80 one-hour sessions (2 to 4 sessions per day within six weeks). From session 1 to session 45, the birds were trained to use the touchscreen as described previously in test 1 (i.e., increasing stepwise the duration before the reward from 5 s to 15 s after the triggering of the music). From session 46 to session 58, the cockatiels were tested under the same modalities as training (condition 1). The pass criterion was 200 pecking occurrences on the screen and 50 pecking occurrences on one given shape for training and condition 1. From session 59 to session 70, the location of shapes changed (condition 2), and from session 71 to the end, the shapes associated with each music play were exchanged (condition 3) (see Table 2 for details).

The following details were recorded for each pecking occurrence:(1)identity of the bird touching the screen(2)touched shape (with the location and the music associated with it)

### 2.2. Statistical Analysis

All the statistical analyses were conducted with R (version 3.6.0 [44]). For each test, we focused on the number of times each individual chose one given shape. First of all, for each individual, the sessions in which less than 3 pecking occurred were not included. Then, for each condition and bird, we evaluated whether, along the sessions, the proportion of choice for a given shape remained stable (i.e., whether there was a session effect or not). For that purpose, per individual and per session, we divided the number of choices for one given shape by the total number of choices (i.e., the total number of pecking occurrences). Then we implemented those proportions of choices in a linear model (LM, function: «lm», package *stats* [44]), for which the *p*-value was calculated thanks to a permutation test (Monte Carlo method with 1000 permutations), using the function «PermTest» from the package *pgirmess* [45]. The permutation test applied to a linear model does not require the residuals to follow a normal distribution, nor does it assume homogeneity of variances. The test randomly redistributes the data to generate an empirical distribution of the statistics under the null hypothesis. This approach allows the observed results to be compared to a theoretical distribution created by the permutations, making the test robust to violations of classical parametric assumptions [46]. Indeed, the statistical unit was the number of sessions per individual and was inferior to 20 for each condition. This model allows us to assess whether individuals are consistent in their choices through the experimentation sessions (i.e., assess a possible session effect). The use of permutation tests on linear models can be performed even if the normality is not respected and there is no residual homoscedasticity. Then, binomial tests were conducted on the total number of choices made per session and per individual (function: «binom.test», package *stats* [44]). Thus, we could evaluate whether, per session, each individual significantly chose a particular shape over the other. In test 2, because the bird Nephtys had chosen the heart shape more often during the last four sessions of condition 1, we conducted a binomial test on all her choices for the last four sessions. The session 8 of condition 1 (test 2) being the only moment when she chose mostly hearts, a binomial test on the last 3 sessions (thus excluding session 8) was also conducted so as to check whether she had changed her preference over time.

## 3. Results

All the figures were designed with the software Power point 365.

### 3.1. Study 1

Seven cockatiels interacted with the device. Among them, four (Gaïa, Nephtys, Éole, and Seth) came regularly and reached the pass criterion (i.e., 300 pecking occurrences on the screen and at least 50 pecking occurrences on one given shape for training and condition 1).

#### 3.1.1. Preferences

Gaïa pecked significantly more at the red square broadcasting rock and roll in conditions 1 and 2. She also pecked significantly more at the dark blue circle broadcasting rock and roll in condition 3. These results suggest that Gaïa had a preference for rock and roll over calm music (see Figure 3 and Table 3 for details).

Like Gaïa, Nephtys pecked significantly more at the red square broadcasting rock and roll in conditions 1 and 2. She also pecked significantly more at the dark blue circle broadcasting rock and roll in condition 3. These results suggest that Nephtys had a preference for rock and roll over calm music (see Figure 4 and Table 3 for details).

Éole pecked significantly more at the dark blue circle broadcasting calm music in conditions 1 and 2. He also pecked significantly more at the red square broadcasting calm music in condition 3. These results suggest that Éole had a preference for calm music over rock and roll (see Figure 5 and Table 3 for details).

Seth pecked significantly more at the dark blue circle in conditions 1, 2, and 3. These results suggest that Seth had a preference for the dark blue circle but no music preferences (see Figure 6 and Table 3 for details).

Three other birds (Odin, Isis, and Hermès) interacted with the screen but did not reach the pass criterion; thus we cannot conclude that they understood how to use the device. Hermès only interacted a few times with the device; as a consequence, no statistical analyses were conducted on his data.

Odin pecked more at the dark blue circle in condition 1 and the red square in conditions 2 and 3, and Isis pecked more at the red square in condition 1 and the blue circle in conditions 2 and 3 (see Table 4 and Table 5).

#### 3.1.2. Cockatiels Progress Across Sessions in the First Study

We found an effect of sessions for Gaïa and Nephtys in condition 1, Seth and Nephtys in condition 2, and Gaïa, Nephtys, and Éole in condition 3 (see Table 5 for details). More precisely, each bird’s preference for a given condition was more and more stable through time. For instance, as mentioned before, Gaïa preferred the red shape broadcasting rock and roll in condition 1. She also pecked significantly more and more often in this shape from the first to the last test sessions in this condition. One exception occurred for Gaïa in condition 2, who seemed to peck significantly less and less at the red shape through time. However, this result could be due to very few atypical sessions displaying extreme values.

### 3.2. Study 2

Two cockatiels (Nephtys and Isis) interacted regularly with the device and reached the pass criterion (see our Section 2).

#### 3.2.1. Preferences

Nephtys pecked significantly more at the green cross in conditions 1, 2, and 3. Those results suggest that Nephtys had a preference for the green cross rather than music preferences (see Figure 7 and Table 5 for details).

Isis pecked significantly more at the orange heart in condition 1 and the green cross in conditions 2 and 3. Those results suggest that Isis had a preference for the left side rather than a certain shape or type of music (see Figure 8 and Table 5 for details).

#### 3.2.2. Cockatiels Progresses Across Sessions in the Second Study

We found an effect of sessions for Nepthys and Isis in condition 1. Like for our first study, it is probably due to the time needed by the birds for learning the new configuration of the screen.

## 4. Discussion

### 4.1. Use of the TouchScreen

Unfortunately, the device did not work as expected. The touchscreen was selected because the grey parrots could successfully use it. Nevertheless, with the cockatiels, either the screen would not activate at all, or the sound would not trigger. This could be due to the smaller size of their beak, which did not press enough surface. This problem highlights the importance of providing a device adapted to the species, both for future studies and for the purpose of enrichment of captive animals’ environments. However, in our studies, several birds were able to learn to peck the shape associated with their preferred music. Thus, with a technically appropriate device, performing choice tests with touchscreens would be possible in this species.

### 4.2. Musical Preferences

Among the birds that interacted sufficiently with the device to meet our pass criterion, three out of four cockatiels exhibited individual musical preferences: Gaïa and Nephtys, two females, preferred rock and roll music over calm music, while Éole, a male, preferred calm music over rock and roll. Given that we exchanged both the locations of the geometric shapes and the shapes associated with each music piece throughout the experiment, these results cannot be explained by a side bias or a preference for a geometric shape or color. Thus, musical preferences exist in this species, at least in some individuals. These results seem in line with the hypothesis according to which vocal learners would be sensitive to musicality: musical preferences have already been shown in some Java sparrows [15] and grey parrots [28]. In grey parrots, those preferences were also different depending on the individuals. These results confirm that Psittaciformes are good candidates for the study of musicality in non-human animals.

Regarding session effects, in most cases, the sessions that differed significantly from the others were the first sessions performed for one given condition. Thus, session effects can be explained by the time the birds need to understand a new configuration of shapes and/or music plays. It is especially true for condition 3 of test 1: Gaïa, Nephtys, and Éole probably needed some trials to get that the music they preferred was not broadcasted by the same shape as before and to adjust their behaviour. Two other effects of sessions (for Nephtys during condition 1 of test 1 and for Isis during condition 1 of test 2) could be related to preferences still being determined: both happened during condition 1, suggesting that the birds might not have established their choice by that time. Yet this seems unlikely for Isis since she had already shown a preference for the left side during test 1 and may have just repeated it during test 2. Another explanation would be that they had not understood yet which shape was associated with which type of music. Two other cases (for Gaïa during condition 2 of study 1 and for Nephtys during condition 1 of study 2) are related to very few atypical sessions displaying extreme values, for which we have no explanation.

Among the other birds who interacted with the device but did not show musical preferences, Odin pecked more on the right side and Isis pecked more on the left side; this may be due to a side preference and/or to a lack of understanding of the device, as they did not reach our pass criterion.

### 4.3. No Preference for Consonance

For the two subjects who interacted with the device during test 2, our results did not show any preference for consonance over dissonance. Indeed, Isis seems to prefer the left side over the right side, rather than the music associated with it. As for Nephtys, in spite of preferring rock and roll over calm music in study 1, in study 2 she seems to prefer the green cross across the three conditions and thus to be indifferent to the consonance/dissonance feature.

Those findings are inconsistent with a universal preference for consonance over dissonance. As it happens, notwithstanding studies supporting a biological preference for consonance in mankind [41] and some other primates [24], contrasted results have been noticed since. Among birds, budgerigars failed to show a preference when tested with a place preference paradigm that allowed them to choose between consonant vs. dissonant music [47]. In humans, 6-month-old infants exposed either to consonant or dissonant music listen more to the type of music they have been exposed to, whether it is consonant or dissonant [48], and members of the Tsimane’ Amazonian tribe, in spite of being able to discriminate consonance from dissonance, do not prefer one type of music over the other [49]. This suggests that the preference for consonant music over dissonant music would also be related to the exposure to such music, and thereby that experience would play a crucial part in this preference. In a previous study (not published yet), our cockatiels showed different behaviour when exposed to consonant versus dissonant music (i.e., less agonistic behaviour during exposure to consonant music as compared to dissonant music). In the current study, the musical stimuli we used were the same as in the previous one, so the birds were exposed to both types of music before this experiment. If familiarity influenced preferences, it could explain why we found a preference for consonance in our previous work but not in this one, as a habituation of the birds to dissonance probably occurred.

### 4.4. Welfare Implications

Our two studies also show that, when provided with such a device, at least some of the birds use it to play their preferred tunes. Although we have limited information about what motivates the birds and how exactly this benefits them, their interaction with the device suggests that it can positively enrich their environment. Captive birds are often kept in small spaces with minimal enrichment. Allowing them to influence their auditory environment by selecting the sounds they prefer could improve their welfare. Indeed, choice and control over the environment have been identified as key factors in animal welfare (see, for example [50]). These possibilities should be explored in further research.

### 4.5. Limitations and Prospects for Further Research

Unfortunately, this study was limited by the device, which did not work as expected. Indeed, as mentioned before, either the screen would not work at all, or the beak of the birds would not trigger the sound. This could be due to the small size of the cockatiels’ beaks, which did not press enough surface. Thereby we could not test whether they would have used the apparatus in the absence of the experimenter, which makes it impossible to conclude whether they pecked preferentially to obtain the music or the food reward. Thus, it would be interesting to provide a device adapted to the birds’ specificities, like in Gupfinger and Kaltenbrunner’s experiment [51], in which grey parrots can interact with an apparatus consisting of rope perches with captors, triggering music when the birds perch on it.

Reproducing our two tests after a delay (of some months or even years) could also give us information about the stability of the birds’ choices: not only would it be interesting for our knowledge, but this would also be important for animal welfare, given that if musical tastes change over time, it would be necessary to adjust sound enrichment depending on their current preferences.

Additionally, in the first experiment, we cannot ensure that birds based their choices on rhythm or other properties of the broadcasted music since the musical features are uncontrolled. This is why the second experiment was based on musical pieces that only differ with the consonance/dissonance feature. We hope that our work will pave the way for other studies, in order to decompose the building blocks of musical preferences.

### 4.6. Vocal Learning Could Be a Continuum

Cockatiels (at least some individuals) seem sensitive to music. Coupled with other data we obtained, such as the positive influence of music on their interactive behaviour (not published yet) and an interest in producing sounds and drumming [52], those results support Watanabe et al.’s hypothesis, which suggests that musical stimuli are mostly reinforcing in species that are capable of vocal learning [20].

Yet, over the past few years, fewer categorical assumptions have been made, arguing for a gradual continuum instead of an absolute jump. First of all, vocal learning might not be a dichotomous feature but a continuum ranging from non-vocal learners and limited vocal learners to complex vocal learners [53]. Additionally, some non-vocal learners display a certain sensitivity to music; for example, a positive effect of music has been evidenced in zoo-housed Western lowland gorillas (*Western lowland gorillas*) [54] and dogs (*Canis lupus familiaris*) [55,56,57]. Other non or limited vocal learners show musical preferences, as is the case of degus (*Octodon degus*), which have highly developed vocal communication and prefer Chilean folk music to Western music and silence [58], or mice, which can develop musical preferences if they are exposed to music during a critical period (between day 15 and day 25 after birth) [59]. These data could be due to a simple enrichment effect, but also to a specific sensitivity to music. Therefore, vocal learning is probably not the only factor linked to musicality in animals.

## 5. Conclusions

Our results seem to show that some cockatiels display individual musical preferences, in line with the idea that, as vocal learners, they would be sensitive to music.

These data could also help to improve animal welfare and pave the way for new research: a motivational theory concerning captive animal welfare suggests that in captive environments, the lack of quantity and diversity of stimuli often results in sensory deprivation, which would induce poor welfare; enriching the senses via, *inter alia*, auditory environmental complexity would reduce negative behaviour [37]. This is why it is important to study independently, as we did, several features of musicality, in order to adapt the best musical broadcast or choices provided for enrichment to captive animals.

Moreover, personal observations showed us that, when we drum against a metal perch, the cockatiels raise their crest and calm down. Therefore, studying more precisely rhythm preferences and influences in those birds could provide further information about their musical sensitivity.

Finally, we showed that cockatiels are able to learn pecking a specific shape among two, on a touchscreen, to have their preferred music displayed. However, a device adapted to their beak size would be necessary to allow them to display the music by themselves without any human intervention.

## Figures and Tables

**Figure 1 animals-14-03609-f001:**
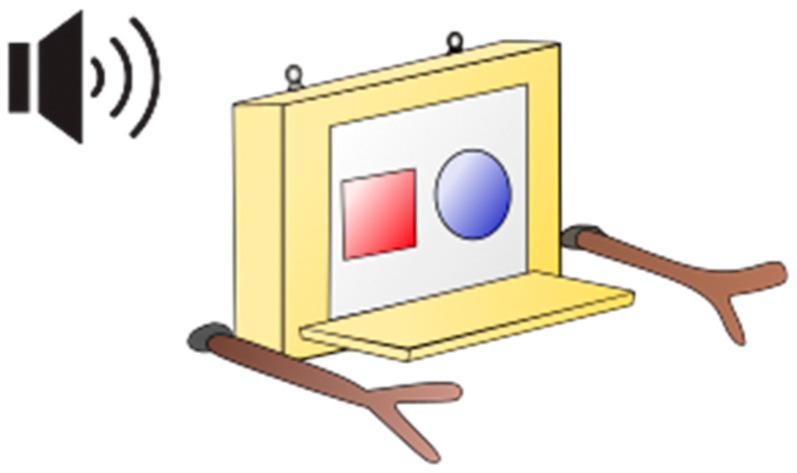
Experimental device.

**Figure 2 animals-14-03609-f002:**
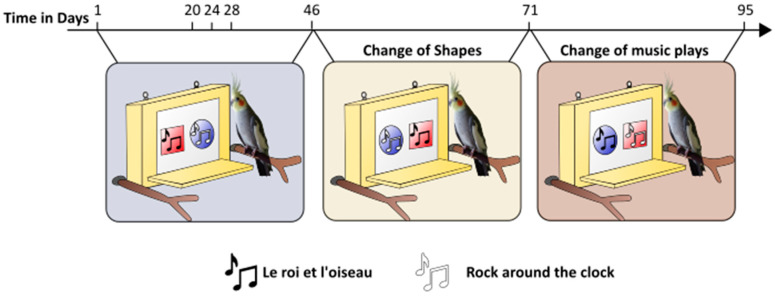
Changes along the sessions during test 1.

**Figure 3 animals-14-03609-f003:**
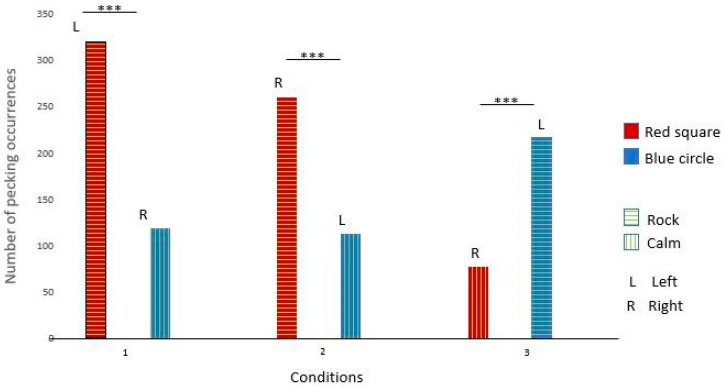
Gaïa’s choices across the three conditions. (***: significant difference)

**Figure 4 animals-14-03609-f004:**
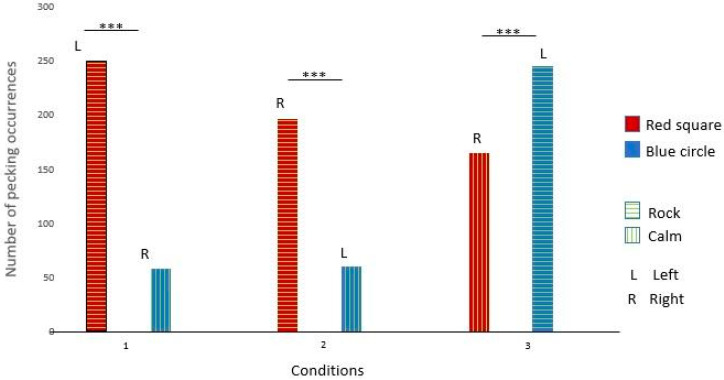
Nephtys’s choices across the three conditions. (***: significant difference)

**Figure 5 animals-14-03609-f005:**
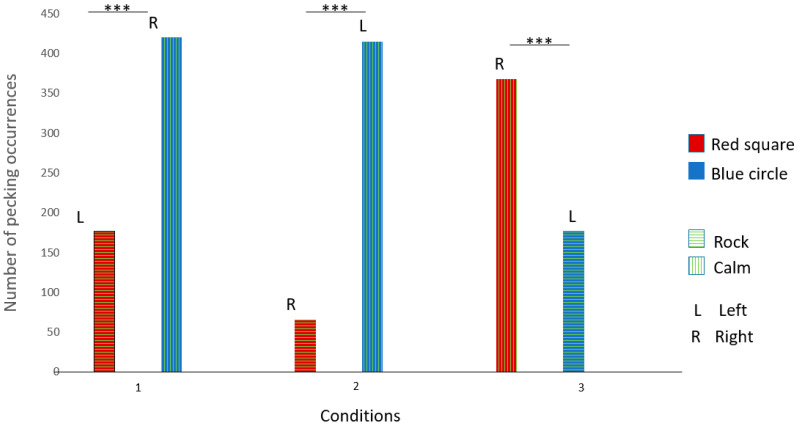
Éole’s choices across the three conditions. (***: significant difference)

**Figure 6 animals-14-03609-f006:**
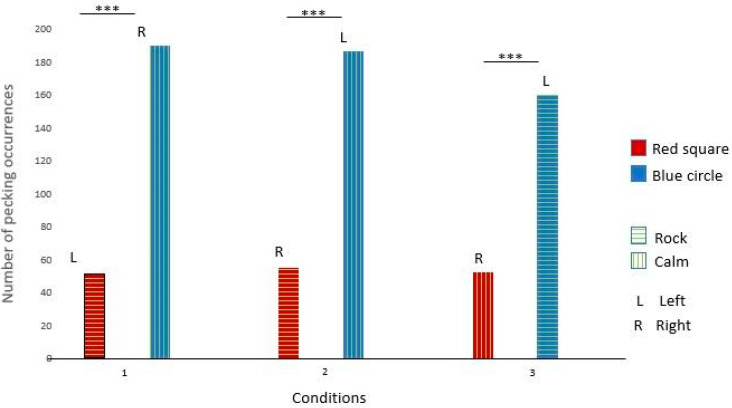
Seth’s choices across the three conditions. (***: significant difference)

**Figure 7 animals-14-03609-f007:**
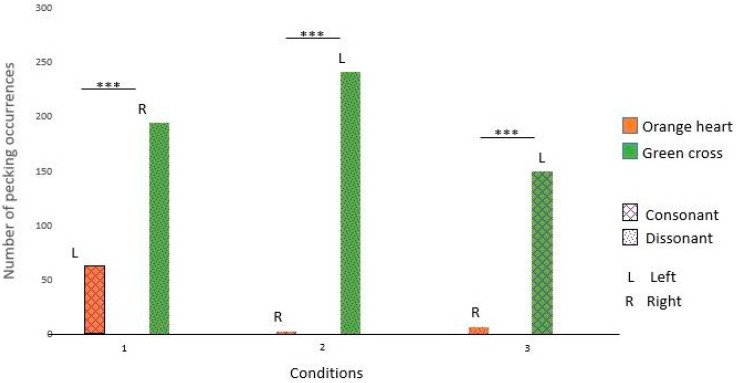
Nephtys’s choices across the three conditions. (***: significant difference)

**Figure 8 animals-14-03609-f008:**
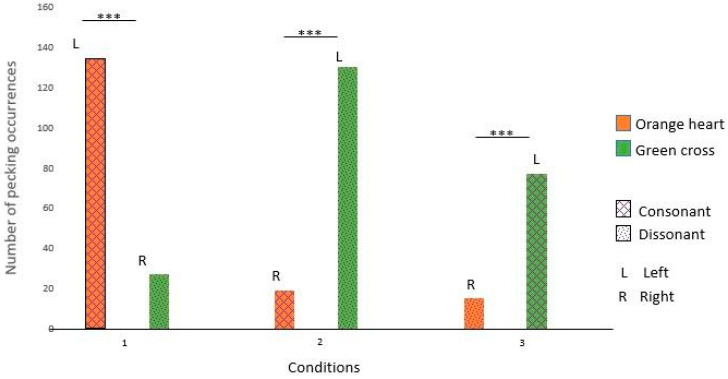
Isis’s choices across the three conditions (***: significant difference)

**Table 1 animals-14-03609-t001:** Name, sex, and age of the birds when each test began.

Bird	Sex	Test 1	Test 2
Hermès	Male	4 years 10 months	5 years 11 months
Callisto	Female	4 years 8 months	5 years 9 months
Viviane	Female	4 years 8 months	5 years 9 months
Nephtys	Female	4 years 9 months	5 years 10 months
Seth	Male	4 years 9 months	5 years 10 months
Skadi	Female	3 years 9 months	4 years 10 months
Odin	Male	3 years 9 months	4 years 10 months
Loki	Male	3 years 9 months	4 years 10 months
Isis	Female	1 year 4 months	2 years 5 months
Éole	Male	1 year 3 months	2 years 4 months
Gaïa	Female	1 year 3 months	2 years 4 months
Morgane	Female	1 year 3 months	2 years 4 months

**Table 2 animals-14-03609-t002:** Types of music, locations, and shapes depending on the condition for each test.

Condition	Shape	Location	Style
Study 1			
1	Red square	Left	Rock and roll
	Dark blue circle	Right	Calm
2	Red square	Right	Rock and roll
	Dark blue circle	Left	Calm
3	Red square	Right	Calm
	Dark blue circle	Left	Rock and roll
Study 2			
1	Orange heart	Left	Consonant
	Green cross	Right	Dissonant
2	Orange heart	Right	Consonant
	Green cross	Left	Dissonant
3	Orange heart	Right	Dissonant
	Green cross	Left	Consonant

**Table 3 animals-14-03609-t003:** Binomial tests and LM with permutation tests depending on conditions and individuals (the significant *p*-values are in bold) for Seth, Gaïa, Nephtys, and Eole during test 1.

Study 1		Binomial Tests	LM with Permutation Test				
Condition	Bird	Success	*p*-Value	Pente (β)	SE	R^2^ Multiple Model	*p*-Value
Condition 1	Seth	0.212	**<0.001**	−0.012	0.008	0.121	0.165
Condition 1	Gaïa	0.728	**<0.001**	0.015	0.005	0.246	**0.011**
Condition 1	Nephtys	0.811	**<0.001**	0.010	0.004	0.333	**0.032**
Condition 1	Eole	0.296	**<0.001**	−0.012	0.009	0.102	0.199
Condition 2	Seth	0.222	**<0.001**	−0.031	0.006	0.613	**<0.001**
Condition 2	Gaïa	0.699	**<0.001**	0.009	0.005	0.144	0.066
Condition 2	Nephtys	0.766	**<0.001**	0.023	0.006	0.456	**0.003**
Condition 2	Eole	0.137	**<0.001**	0.002	0.003	0.025	0.46
Condition 3	Seth	0.245	**<0.001**	−0.009	0.008	0.073	0.241
Condition 3	Gaïa	0.262	**<0.001**	−0.016	0.007	0.202	**0.028**
Condition 3	Nephtys	0.402	**<0.001**	−0.031	0.007	0.461	**<0.001**
Condition 3	Eole	0.675	**<0.001**	0.013	0.004	0.295	**0.001**
**Study 2**		**Binomial tests**	**LM with permutation test**				
**Condition**	**Bird**	**Success**	***p*-value**	**Slope (β)**	**SE**	**R^2^ Multiple Model**	***p*-value**
Condition 1	Nephtys	0.242	**<0.001**	−0.053	0.014	0.616	**0.002**
Condition 1	Isis	0.832	**<0.001**	0.075	0.031	0.427	**0.047**
Condition 2	Nephtys	0.008	**<0.001**	0.006	0.004	0.268	0.212
Condition 2	Isis	0.128	**<0.001**	0.008	0.014	0.047	0.583
Condition 3	Nephtys	0.039	**<0.001**	−0.009	0.013	0.071	0.577
Condition 3	Isis	0.163	**<0.001**	−0.001	0.031	3.069 × 10^−5^	0.987

**Table 4 animals-14-03609-t004:** Pecking occurrences for Odin and Isis during test 1.

Condition	Bird	Red Square	Blue Circle
1	Odin	49	165
1	Isis	71	21
2	Odin	50	29
2	Isis	27	93
3	Odin	15	5
2	Isis	3	46

**Table 5 animals-14-03609-t005:** Binomial tests and LM with permutation tests for Isis and Odin during study 1.

Test 1		Binomial Tests		LM with Permutation Test			
Condition	Bird	Success	*p*-Value	Slope (β)	SE	R^2^ Multiple Model	*p*-Value
Condition 1	Odin	0.229	**<0.001**	0.03719	0.03088	0.1535	0.281
Condition 1	Isis	0.772	**<0.001**	−0.04059	0.04509	0.1684	0.355
Condition 2	Odin	0.633	**0.01191**	0.11228	0.04541	0.5501	**0.046**
Condition 2	Isis	0.225	**<0.001**	0.01126	0.01415	0.05009	0.469
Condition 3	Odin	0.75	**0.02069**	−0.16667	0.06706	0.6069	0.06
Condition 3	Isis	0.061	**<0.001**	−0.02857	0.02073	0.1472	0186

## Data Availability

The data are available at the following link: https://osf.io/z2ykw/?view_only=3853a917b4aa46958b3834d93a410ce0 (accessed on 9 December 2024).

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
