# Peer review of "Do Cockatiels Choose Their Favourite Tunes? Use of Touchscreen for Animal Welfare Enhancement and Insights into Musical Preferences"

_animals, 2024, doi:10.3390/ani14243609_

Round 1
Reviewer 1 Report
Comments and Suggestions for Authors
The topics approached are very interesting, the introduction is very well documented, also the study's secondary goal novelty is obvious - approaching a niche topic (relationship between musical preferences and personality in non-human animals). Moreover, tracking the problem of cockatiels control over their auditory environment by choosing music tracks on touch screens may bring some benefit in the future for enhancing the welfare in such a vocal learner species, via auditory EE procedures.
However, some inconsistencies can be noticed and some aspects require real improvement:
- Please give more details about the subject of the study and the experimental protocol in relation with the expected results. You have to explain more clear why the conditions are suffering major changes between study 1 and study 2 (color and shape of simulation buttons in study 1 - real buttons on touchscreen in study 2, different music tracks for dissonant-consonant) and why the period between the two tests was 11 months (maybe for an in depth insight on the particular personality traits in the cockatiels - if I understood right, for both studies the same group of birds was used, or for revealing some findings on long-term memory in these birds, or for other reason). On page 17 is stated that: "In a previous study, our cockatiels showed different behavior when exposed to consonant versus dissonant music. In the current study, the musical stimuli we used were the same as in the previous one, therefore the birds were exposed to both types of music before this experiment and may be habituated to them." If the group of birds for the present study was the same as you used previously, maybe the habituation to both type of music (consonant, dissonant) can represent a bias in the birds responses? Please specify clearer the previous experiences of the birds with the music stimuli and the aims of your studies. Also, it is very unfortunate that the selected touch screen doesn't have the adequate sensibility in relation with the beak size of the birds (the implication of the examiner in the process could be again a bias source). Maybe a more inspired selection of a device in the future or connect them with rope perches with captors, as you remind being used in Kaltenbrunner’s experiment in 2017 can solve such issues in the future research on this fascinating topic.
You should indicate clearly (in the chapter Material and methods, on page 4) the origin of the birds and also the fact that they were kept under similar housing/micro-climate conditions, prior and during the experiments or for their entire life. They are most likely animals kept for scientific purpose - in your ethology studies facility, belonging to specific genetic lines. Moreover, giving some additional details about the housing conditions (in terms of respecting the optimal microclimate parameters) and the lighting program applied will be useful: the transition light to dark and vice-versa were done gradually (dimmable light sources), during the night you applied any lighting, it is any bird showing night frights?
On page 4, you stated that the experiment was held at your laboratory with twelve cockatiels. But in table 7 on page 14 there are 13 birds (a supplementary cockatiel called Bahloo).
- It is better to not mention commercial names, brands e.g., (replace them to available on the market: Ryght Exago, Deli Nature, ZuPreem etc.)
- in the last paragraph of the abstract, use somewhere the keyword environmental enrichment (EE), music is considered in some cases an EE procedure for bird’s welfare level enhancement. Of course, you mention this in chapter 5. Conclusion, but try it too in the abstract. In the abstract the term rock is used, but it is a very broad genera, including many categories (rock ballads - which are consonant?, electronic music, grunge, heavy, hard, punk etc.) - you can replace it with rough music / more rhythmical? In material and methods, the first track, Bill Haylay song, is not a rock - but a rock-and-roll piece. Mention the similarities between the two songs in study 1 in relation to the other two in study 2. Also, specify which was the criteria for selecting the particular songs - that you were advised by a musician professional.
- some typos must be solved, some content must be rephased: orang-utans (hyphening), Paper screen was only visible when ... - replace it with ...visible to the birds when, Food was removed thirty minutes before each session, to motivate them for alimentary rewards... - replace it with In order to increase the motivation for the alimentary rewards offered during the sessions' periods, the food was removed thirty minutes before such accustoming trials (otherwise the sentence seems somehow contradictory), The two music plays broadcasted were Rock around the clock by Bill Haylay and his comets (rock music) and Le roi et l’oiseau (credit movie) - maybe can be replaced with ... comets (rock and roll song) ..... a movie credit calm soundtrack, location of shapes was exchanged to check - was switched to check, had music preferences or only preferences for one shape or another - .... or only preferences for a certain shape, Two Raters evaluated twice each .... two raters (lowercase)..., Per session, each individual chose significantly more one shape than the other - ... each individual significantly chose a particular shape over the other, Thus, we cannot conclude that they understood the device - ... understood how to use the device,
Maybe the titles 3.1.2 and 3.2.2. Effect of sessions can be renamed (Cockatiels progress during the sessions of the first study / and second study, respectively)
table 6 on page 13 IrriTable - lowercase T (Irritable)
on page 14 don't forget about the new bird Bahloo (it is included in the group or not? - and also the results for this particular bird)
- the references were written in a rush, some titles are not present, some numbering on the references are not correct or continue for the same entry on the next row etc. (reference 16 - the title and the data for the article are missing, reference 18-19, reference 22-23, 24-25, blank space 35-36, reference 47, 53, 81 etc.)
Solve the issues at the best possible manner, the results and the perspectives are really valuable!
Author Response
Please give more details about the subject of the study and the experimental protocol in relation with the expected results. You have to explain more clear why the conditions are suffering major changes between study 1 and study 2 (color and shape of simulation buttons in study 1 - real buttons on touchscreen in study 2, different music tracks for dissonant-consonant) and why the period between the two tests was 11 months (maybe for an in depth insight on the particular personality traits in the cockatiels - if I understood right, for both studies the same group of birds was used, or for revealing some findings on long-term memory in these birds, or for other reason).
Response : Thank you for pointing this out. We agree with this comment. We added some précisions in the manuscript.
On page 17 is stated that: "In a previous study, our cockatiels showed different behavior when exposed to consonant versus dissonant music. In the current study, the musical stimuli we used were the same as in the previous one, therefore the birds were exposed to both types of music before this experiment and may be habituated to them." If the group of birds for the present study was the same as you used previously, maybe the habituation to both type of music (consonant, dissonant) can represent a bias in the birds responses? Please specify clearer the previous experiences of the birds with the music stimuli and the aims of your studies. Also, it is very unfortunate that the selected touch screen doesn't have the adequate sensibility in relation with the beak size of the birds (the implication of the examiner in the process could be again a bias source). Maybe a more inspired selection of a device in the future or connect them with rope perches with captors, as you remind being used in Kaltenbrunner’s experiment in 2017 can solve such issues in the future research on this fascinating topic.
Response : Thank you for pointing this out. We agree with this comment. We added some precisions in the discussion.
You should indicate clearly (in the chapter Material and methods, on page 4) the origin of the birds and also the fact that they were kept under similar housing/micro-climate conditions, prior and during the experiments or for their entire life. They are most likely animals kept for scientific purpose - in your ethology studies facility, belonging to specific genetic lines. Moreover, giving some additional details about the housing conditions (in terms of respecting the optimal microclimate parameters) and the lighting program applied will be useful: the transition light to dark and vice-versa were done gradually (dimmable light sources), during the night you applied any lighting, it is any bird showing night frights?
Response : Thank you for pointing this out. We agree with this comment. We added some precisions in the chapter Material and methods.
On page 4, you stated that the experiment was held at your laboratory with twelve cockatiels. But in table 7 on page 14 there are 13 birds (a supplementary cockatiel called Bahloo).
Response : Thank you for pointing this out. We agree with this comment. Mentioning Bahloo was a mistake since he was not included in the experiment. We thus deleted him from the study, and from the manuscript.
It is better to not mention commercial names, brands e.g., (replace them to available on the market: Ryght Exago, Deli Nature, ZuPreem etc.)
Response : Thank you for pointing this out. We agree with this comment. We corrected as suggested.
In the last paragraph of the abstract, use somewhere the keyword environmental enrichment (EE), music is considered in some cases an EE procedure for bird’s welfare level enhancement. Of course, you mention this in chapter 5. Conclusion, but try it too in the abstract. In the abstract the term rock is used, but it is a very broad genera, including many categories (rock ballads - which are consonant?, electronic music, grunge, heavy, hard, punk etc.) - you can replace it with rough music / more rhythmical? In material and methods, the first track, Bill Haylay song, is not a rock - but a rock-and-roll piece. Mention the similarities between the two songs in study 1 in relation to the other two in study 2. Also, specify which was the criteria for selecting the particular songs - that you were advised by a musician professional.
Response. Thank you for pointing this out. We agree with this comment. We clarified as suggested by using the keyword EE and replaced "rock" by "rock and roll".
Some typos must be solved, some content must be rephased: orang-utans (hyphening), Paper screen was only visible when ... - replace it with ...visible to the birds when, Food was removed thirty minutes before each session, to motivate them for alimentary rewards... - replace it with In order to increase the motivation for the alimentary rewards offered during the sessions' periods, the food was removed thirty minutes before such accustoming trials (otherwise the sentence seems somehow contradictory), The two music plays broadcasted were Rock around the clock by Bill Haylay and his comets (rock music) and Le roi et l’oiseau (credit movie) - maybe can be replaced with ... comets (rock and roll song) ..... a movie credit calm soundtrack, location of shapes was exchanged to check - was switched to check, had music preferences or only preferences for one shape or another - .... or only preferences for a certain shape, Two Raters evaluated twice each .... two raters (lowercase)..., Per session, each individual chose significantly more one shape than the other - ... each individual significantly chose a particular shape over the other, Thus, we cannot conclude that they understood the device - ... understood how to use the device,
Maybe the titles 3.1.2 and 3.2.2. Effect of sessions can be renamed (Cockatiels progress during the sessions of the first study / and second study, respectively)
table 6 on page 13 IrriTable - lowercase T (Irritable)
on page 14 don't forget about the new bird Bahloo (it is included in the group or not? - and also the results for this particular bird)
- the references were written in a rush, some titles are not present, some numbering on the references are not correct or continue for the same entry on the next row etc. (reference 16 - the title and the data for the article are missing, reference 18-19, reference 22-23, 24-25, blank space 35-36, reference 47, 53, 81 etc.)
Response : Thank you for pointing this out. We agree with those comments. We modified everything as suggested.
Reviewer 2 Report
Comments and Suggestions for Authors
In my opinion this is a properly written manuscript about an interesting study. The scope on Ethology with strong animal welfare implications makes the manuscript even more interesting.
However, I second the decision of the authors to leave it with the hint to the animal welfare implications with only a rather short mentioning in the discussion. This seems appropriate due to the fact that the sample size of those birds really delivering sound data for analysis and interpretation became quite small in the study. Thus there is a strong hint towards a possible and likely beneficial application of the setup in the context of enrichment to increase animal welfare in captivity but it seems too early for sound statements about this given the limited sample size.
In my opinion the wider setting of the study as well as methods and results are presented clearly and in an appropriate way and I particularly welcome the honest discussion of limitations the authors faced during the study. Besides a few very minor things to amend (see below) I recommend publication.
Some minor suggestions:
chapter 1.2, paragraph 2: please revise the list of taxa with shown abilities to discriminate rhythms or melodies: the starling is in there twice with two different citations and it might generally make sense to give the list an alphabetic, chronological or taxonomic order. Further down in this paragraph, it is not necessary to add the same citation each time you mention the species. This will help to make this listing easier to read.
page 2, 6 lines from bottom: the composer spells Schönberg or, if this letter is not available, can be spelled as Schoenberg. But not Schöenberg.
chapter 2.1.1: if the birds had to listen to the end of the music before being rewarded they might learn to go always for the shorter piece of music. The only info on duration is "i.e. 15 seconds". Does that mean there were differences in length between different pieces and how was this accounted for?
Chapter 3.1.2: please make sure the bird names are consistently spelled the same: Gaïa / Gaia
Author Response
chapter 1.2, paragraph 2: please revise the list of taxa with shown abilities to discriminate rhythms or melodies: the starling is in there twice with two different citations and it might generally make sense to give the list an alphabetic, chronological or taxonomic order. Further down in this paragraph, it is not necessary to add the same citation each time you mention the species. This will help to make this listing easier to read.
Response : Thank you for pointing this out. We agree with this comment. We clarified as suggested.
page 2, 6 lines from bottom: the composer spells Schönberg or, if this letter is not available, can be spelled as Schoenberg. But not Schöenberg.
Response : Thank you for pointing this out. We agree with this comment. We modified as suggested.
chapter 2.1.1: if the birds had to listen to the end of the music before being rewarded they might learn to go always for the shorter piece of music. The only info on duration is "i.e. 15 seconds". Does that mean there were differences in length between different pieces and how was this accounted for?
Response : Thank you for pointing this out. We agree with this comment. We clarified as suggested.
Chapter 3.1.2: please make sure the bird names are consistently spelled the same: Gaïa / Gaia
Response : Thank you for pointing this out. We agree with this comment. We modified as suggested.
Reviewer 3 Report
Comments and Suggestions for Authors
I recommend reviewing the section dedicated to the analysis of the personality of the subjects investigated. It suffers from a lack of balance and strength from a scientific point of view.
Author Response
Thank you for your suggestion. We agree with you that we have not enough data and subjects to keep our section on personality. Therefore we deleted all parts liked to this theme.
Reviewer 4 Report
Comments and Suggestions for Authors Dear Editor, The aim of this exploratory study was to verify whether cockatiels can learn to use a touchscreen to select their preferred type of music from two choices. In addition, the authors aimed to relate the birds' personality type to their musical preferences. The results were mixed. Of the 12 cockatiels tested, three learned, after numerous trials, to discriminate the shapes (squares and circles) associated with different musical extracts. Moreover, the choice of music varied from bird to bird. Two birds selected one type of music as a priority, while the other bird preferred another type of music. As for the attempt to link personality and musical preferences, as the authors mention, the results are difficult to interpret due to the limited data available. In the discussion, the authors explore various alternatives (adapting the device to the cockatiels' beaks, etc.) that could help to better understand cockatiels' musical choices. Here is my overall assessment. I liked the objective of the study and believe that this type of study is necessary to improve housing conditions for captive animals. In addition, I appreciated the statistical approaches used (permutation tests and principal component analyses). Nevertheless, several elements of the manuscript prompt me to propose a major revision. Here are the main reasons:- The introduction is relatively difficult to grasp. I felt like I was jumping from one point to another without a common thread, mainly in section 1.1 and the first paragraph of section 1.2. For readers of Animals, I have the impression that the text begins in earnest only in the second paragraph of section 1.2. In my opinion, the concepts listed above (musicality, emotion, personality, etc.) could be introduced from this section without devoting a specific section to them.
- All sections (introduction, methodology, results, discussion) dealing with the personality of birds in relation to their musical preferences should be removed from the manuscript. This theme considerably weakens the manuscript and disperses the reader's interest from the main objective, which is to test whether birds are able to learn to select a specific type of music using a touchscreen device.
- Do the results obtained for 3 of the 12 birds mean that they prefer one type of music to another? I doubt it. In fact, the present study reminds me of Raynolds' study (1961) which demonstrated that pigeons are able to discriminate a geometric shape (circle, triangle) to obtain food. In the present study, 3 of the 12 birds learned to discriminate a circle or square to obtain reinforcement. In addition, as in the present study, Raynolds' (1961) pigeons showed preferences. One pigeon (#107) preferred the circle and one pigeon (#105) preferred the triangle. In the present study, we observe the same phenomenon except that the choice of shape was associated with specific music.
- This raises the question of whether the presence of music following a choice (square or circle) is really associated with the musical choice made by the birds [here, the authors hypothesize, without proof, that the birds associated the specific shape with the music]. Personally, I believe that the delay in reinforcement, following the choice of the specific shape, led to a delay in learning. Moreover, delaying the reinforcement delay is well known to decrease the speed of learning in operant conditioning (see Harker, 1950). So, if instead of associating a musical extract with a shape, the authors had associated a video extract, it's possible that the same results would have been observed. This last point is extremely important and must be addressed by the authors in their revision of the manuscript.
Author Response
Thank you for your comments and suggestions.
We deleted the section and 1.1 and rewrote the section 1.2 (which is now 1.1) according to your suggestions.
We also deleted all sections dealing with the personality of birds in relation to their musical preferences.
Following your advice, we looked at Reynolds’ (1961) interesting paper. In this paper, two pigeons were conditioned to respond to a white triangle with a red background, but not to a white circle on a green background. Afterward, when the triangle and the red color were presented separately, only the presentations of the triangle for Pigeon 105, and only the presentations of a red key for Pigeon 107, resulted in responding. Therefore, one pigeon attended to the red color and the other to the triangle, even though the responding of each had been reinforced in the presence of the triangle superimposed on the red key.
We agree that some birds could prefer a geometric shape over another. It is precisely our interpretation of the results obtained by one of the birds, as we wrote in the paper : “These results suggest that Seth had a preference for the dark blue circle but no music preferences”.
However, we took care to design our study in a way that allowed us to differentiate between a preference for a geometric shape (or a color), or a side preference, and a preference for a musical piece over another : that is why, across the experiment, the location of shapes changed (to control for a side preference : condition 2) and then the shapes associated with each music piece were exchanged (condition 3: to control for a preference for a geometric shape or a color). If they had a preference for a geometrical shape, Gaïa, Nephtys and Éole would keep choosing the same shape in condition 3. We clarified this in the Methods and in the Discussion.
Concerning, the delay in reinforcement, it is true that it could decrease the speed of learning. However, there was no delay between the peck on the screen and the beginning of the music (only a delay between the peck and the food reward) ; therefore, our procedure may have delayed the association between the choice of a shape and the food reward, but not the association between the choice of a shape and the music. We introduced the delay because we wanted the birds to listen to the music before being focused on eating the food.
We agree that if instead of associating a musical extract with a shape, we had associated a video extract, it's possible that the same results would have been observed : but here, there was no video extract, and although there was a visual stimuli (the geometric shape), the choice of the three birds (Gaïa, Nephtys and Éole) who in our opinion showed musical preferences, were consistent with a preference for a musical piece (which was always preferred across our three conditions) and not for the geometrical shape (that was not the same preferred in condition 1 and 2 vs. condition 3).
Round 2
Reviewer 1 Report
Comments and Suggestions for Authors
The authors succeed to clarify their goals in their experimental protocol included in the paper on such a fascinating topic. Their explanations and correction are reasonable. Moreover, in the present version they use in a larger extent the proposed EE term and they also present some additional connections with welfare topic, which would be beneficial in terms of article's relevance.
However, there are some minor issues, highlighted by me in the green sections of the attached .pdf file, for instance:
- use of rock (instead of proposed rock and roll, as rock is a very broad term: in the simple summary, in the chapters with the Material and Methods, Results)
- some commercial brands are still mentioned
- it is not clear the relation of the paragraph below the table 2 with the table content (and the numbering: 5. and 6.)
- in two places the session number must be corrected (on page 6, before figure 2 is session 71? and on page 7 before Table 2 is session no. 59?)
- there are still some alignment issues/inconsistencies in the References chapter
Please solve these problems in an appropriate manner.

Author Response
We modified the manuscript as suggested in the comment, and highlighted the changes in yellow.
Reviewer 4 Report
Comments and Suggestions for Authors
I have no further comment. The authors have removed the section on personality traits and musical preferences, which makes this manuscript much easier to read. The manuscript is now easy to read and the conclusions are limited to the data observed by the authors.